plant science, environmental science, evolution

vascular plants, leaf mesophyll, intercellular airspace, gas diffusion

**Authors for correspondence:**
Guillaume Théroux-Rancourt
e-mail: guillaume.theroux-rancourt@boku.ac.at
Adam B. Roddy
e-mail: aroddy@fiu.edu

†These authors contributed equally to this study.

# Maximum CO₂ diffusion inside leaves is limited by the scaling of cell size and genome size

Guillaume Théroux-Rancourt[1,†], Adam B. Roddy[2,†],
J. Mason Earles[3,4], Matthew E. Gilbert[5], Maciej A. Zwieniecki[5],
C. Kevin Boyce[6], Danny Tholen[1], Andrew J. McElrone[3,7], Kevin A. Simonin[8]
and Craig R. Brodersen[9]

[1]Institute of Botany, University of Natural Resources and Life Sciences, 1180 Vienna, Austria
[2]Institute of Environment, Department of Biological Sciences, Florida International University, Miami, FL 33199, USA
[3]Department of Viticulture and Enology, [4]Department of Biological and Agricultural Engineering, and [5]Department of Plant Sciences, University of California, Davis, CA 95616, USA
[6]Department of Geological Sciences, Stanford University, Palo Alto, CA 94305, USA
[7]USDA-Agricultural Research Service, Davis, CA 95616, USA
[8]Department of Biology, San Francisco State University, San Francisco, CA 94132, USA
[9]School of the Environment, Yale University, New Haven, CT 06511, USA

GT-R, 0000-0002-2591-0524; ABR, 0000-0002-4423-8729; JME, 0000-0002-8345-9671; MEG, 0000-0002-6761-7975; MAZ, 0000-0002-3774-4455; CKB, 0000-0002-3980-6066; DT, 0000-0002-9517-0939; AJM, 0000-0001-9466-4761; KAS, 0000-0002-4990-580X; CRB, 0000-0002-0924-2570

Maintaining high rates of photosynthesis in leaves requires efficient movement of CO₂ from the atmosphere to the mesophyll cells inside the leaf where CO₂ is converted into sugar. CO₂ diffusion inside the leaf depends directly on the structure of the mesophyll cells and their surrounding airspace, which have been difficult to characterize because of their inherently three-dimensional organization. Yet faster CO₂ diffusion inside the leaf was probably critical in elevating rates of photosynthesis that occurred among angiosperm lineages. Here we characterize the three-dimensional surface area of the leaf mesophyll across vascular plants. We show that genome size determines the sizes and packing densities of cells in all leaf tissues and that smaller cells enable more mesophyll surface area to be packed into the leaf volume, facilitating higher CO₂ diffusion. Measurements and modelling revealed that the spongy mesophyll layer better facilitates gaseous phase diffusion while the palisade mesophyll layer better facilitates liquid-phase diffusion. Our results demonstrate that genome downsizing among the angiosperms was critical to restructuring the entire pathway of CO₂ diffusion into and through the leaf, maintaining high rates of CO₂ supply to the leaf mesophyll despite declining atmospheric CO₂ levels during the Cretaceous.

## 1. Introduction

The primary limiting enzyme in photosynthesis, rubisco, functions poorly under low CO₂ concentrations. For leaves to sustain high rates of photosynthesis, they must maintain high rates of CO₂ supply from the atmosphere to the sites of carboxylation in the leaf mesophyll. The importance of maintaining efficient CO₂ diffusion into the leaf is reflected in the evolutionary history of leaf anatomy; leaf surface conductance has increased during periods of declining atmospheric CO₂ concentration [1], primarily due to increasing the density and reducing the sizes of stomatal guard cells that form the pores in the epidermis through

which $CO_2$ diffuses [2–5]. However, allowing $CO_2$ to diffuse into the leaf exposes the wet internal leaf surfaces to a dry atmosphere. Therefore, maintaining a high rate of $CO_2$ uptake necessarily requires high fluxes of water to be delivered throughout the leaf to replace water lost during transpiration (electronic supplementary material, figure S1), which is accomplished by a dense network of veins [6,7]. Coordinated increases in the densities of leaf veins and stomata, and reductions in stomatal guard cell size, enabled the elevated photosynthetic rates that occurred only among angiosperm lineages despite declining atmospheric $CO_2$ concentration during the Cretaceous [1,5,8–13].

For a given leaf volume, the number of cells that can be packed into a space and the distance between different cell types is fundamentally limited by the size of these cells [12,14]. Because cells occupy physical space and increasing investment in any one cell type will displace other cell types [15,16], reducing cell size is hypothesized to be the primary way of allowing more cell types and more cell surface area of a given type to be packed into a given leaf volume. Thus, factors that limit the minimum size of cells represent fundamental constraints on the cellular organization of leaves. While numerous environmental, physiological and genetic factors can influence the final sizes of somatic cells, the minimum size of a cell is limited by the volume of its nucleus, which is commonly measured as genome size [17–20]. Experimental tests of the effects of genome size on cell size have shown that doubling genome size by arresting mitosis results in larger and less abundant stomata and mesophyll cells [20–22]. Reductions in cell size and increases in cell packing densities that occurred for veins and stomata only among angiosperm lineages therefore required reductions in genome size [13]. While reducing cell size and increasing cell packing density elevate maximum stomatal conductance to $CO_2$ [4,13], realizing the potential benefits of elevated stomatal conductance to $CO_2$ diffusion would require modifications to the internal leaf structure that most limits $CO_2$ transport: the absorptive mesophyll cell surface area exposed to the intercellular airspace.

Diffusion of $CO_2$ inside the leaf is a major limitation to photosynthesis [23,24] and has been considered to be a prime target for selection to increase photosynthetic capacity [25]. Unlike other tissues, the mesophyll is defined by its intercellular airspace as much as by the cells themselves, both of which determine the overall $CO_2$ conductance of the tissue. The conductance of the intercellular airspace ($g_{ias}$) is thought to be much higher than the liquid-phase conductance ($g_{liq}$) through the cell walls, cell membranes, and into the chloroplast stroma [26,27] because $CO_2$ diffusivity is approximately 10 000 times higher in air than in water. These two conductances are arranged roughly in series, with $g_{liq}$ acting as a greater limitation to $CO_2$ uptake. While multiple membrane [24] and intracellular factors, such as carbonic anhydrase activity [28] and chloroplast positioning [29], can be actively controlled to rapidly change $g_{liq}$ over short timescales, once a leaf is fully expanded, the structural determinants of $g_{ias}$ and $g_{liq}$, which include the sizes and configurations of cells and airspace in the mesophyll, are thought to be relatively fixed [24,25,30]. Of the various structural determinants of $g_{liq}$ [30], the three-dimensional (3D) surface area of the mesophyll exposed to the intercellular airspace ($SA_{mes}$) is thought to be the most important because it defines the maximum amount of cell surface area that chloroplasts can occupy [26,27]. Because

variation in leaf and mesophyll thicknesses influences $SA_{mes}$ per leaf area [31], expressing $SA_{mes}$ instead by tissue volume ($V_{mes}$, i.e. the sum of the mesophyll cell volume, $V_{cell}$, and the airspace volume, $V_{air}$) accounts for variation in leaf construction [32,33]. The surface area of the mesophyll per tissue volume ($SA_{mes}/V_{mes}$; electronic supplementary material, figure S2), therefore, is the primary tissue-level structural trait limiting $CO_2$ diffusion from the intercellular airspace into the hydrated cell walls of the mesophyll.

Because smaller cells have a higher surface area per volume than larger cells, reducing cell size by genome downsizing would allow for more surface area per cell volume ($SA_{cell}/V_{cell}$) and per total tissue volume ($SA_{mes}/V_{mes}$) that would result in an increase in available diffusive area and the potential for higher rates of $CO_2$ supply to the chloroplasts. We hypothesized that cell sizes and packing densities of all cell types in a leaf are fundamentally constrained by genome size [4,5,12,13,19–21,34]. Specifically, we predicted that genome size limits minimum cell size such that smaller genomes allow for a larger range of final cell size in tissues throughout the leaf. Similarly, because more cells can be packed into a given space if these cells are smaller, we predicted that smaller genomes would also allow for higher cell packing densities and greater variation in cell packing densities. Thus, we predicted that the simple requirement that a cell contain its genome would affect cell sizes and cell packing densities of all cell types in the leaf, thereby influencing tissue-level structure and function. In this way, genome downsizing was predicted to allow for smaller cells and higher cell packing densities not only of veins and stomata but also in the mesophyll. The elevated $SA_{mes}/V_{mes}$ enabled by smaller mesophyll cells is predicted to have been an essential innovation among early angiosperms that enabled their elevated rates of $CO_2$ supply to the photosynthesizing mesophyll cells despite declining atmospheric $CO_2$ concentrations during the Cretaceous [1,5,8–11,13,20,35,36].

We tested these hypotheses using high resolution, 3D X-ray microcomputed tomography (microCT) to characterize cell sizes, cell packing densities and the exposed 3D surface area of the mesophyll tissue of leaves spanning the extant diversity of vascular plants (electronic supplementary material, table S1). To test how these anatomical innovations in the leaf mesophyll influence $CO_2$ diffusion, we modelled $g_{ias}$ and $g_{liq}$ as a function of cell size and porosity. The mesophyll tissue of most leaves is composed of two distinct layers, the palisade and the spongy mesophyll, which are thought to be optimized for different functions [37,38]. We analysed these two layers separately to determine how differences in their 3D tissue structure (electronic supplementary material, figures S1 and S2) may drive differences in $g_{ias}$ and $g_{liq}$.

## 2. Results and discussion

### (a) Genome downsizing enables re-organization of the leaf mesophyll

For 86 species spanning the extant diversity of vascular plants (electronic supplementary material, table S1), we quantified from microCT images the sizes of spongy and palisade mesophyll cells and stomatal guard cells, as well as the packing densities per unit leaf area of veins, stomata and palisade mesophyll cells. We first tested whether genome size limited

**Figure 1.** (*a*,*b*) Cell volumes, (*c*,*d*) cell packing densities, and (*e*,*f*) total mesophyll surface area per tissue volume ($SA_{mes}/V_{mes}$) in leaves scale with 2C genome size across vascular plants (angiosperms, blue; gymnosperms, orange; ferns and fern allies, grey). Minimum cell volumes (modelled from cell diameters) and maximum cell packing densities are limited by the size of meristematic cells (solid lines). Measurements of meristematic cells as a function of genome size in log-log space (*b*, solid line; from [19]) are reproduced in arithmetic space (*a*). Theoretical maximum packing density of meristematic cells (*c*,*d*) was calculated from measured cell volumes [19] as the reciprocal of meristematic cell cross-sectional area (see Material and methods) assuming spherically shaped cells.

the volumes and packing densities of stomatal guard cells and palisade mesophyll cells by comparing them to published measurements of meristematic cell volume as a function of genome size (figure 1) [19]. The shapes of palisade mesophyll cells and stomatal guard cells can be approximated as capsules, such that cell volumes can be calculated from linear dimensions of length or diameter (see Material and methods) [20,39]. Mature plant cells are always larger than their meristematic precursors, often considerably larger (figure 1*a*,*b*) [19–21,34]. By reducing the size of meristematic cells, genome downsizing allows for smaller minimum cell size and also a greater range in mature cell size of both stomatal guard cells and palisade mesophyll cells (figure 1*a*), consistent with prior results [13,20]. These effects of genome size on cell size were also reflected in the packing densities of guard cells and palisade mesophyll cells (figure 1*c*,*d*). Smaller genomes raised the upper limit on maximum packing densities of meristematic cells, allowing for higher packing densities of both guard cells ($D_{stom}$) and palisade mesophyll cells ($D_{palisade}$), consistent with prior results for veins, stomata [13,22] and mesophyll cells [21,34]. Not only did smaller genomes result in smaller cells and higher cell packing densities, but smaller genomes also allowed for greater variation in cell sizes and

cell packing densities of stomata, mesophyll and veins (figure 1*a*,*c*; electronic supplementary material, figure S3) [13,20,40]. While the shapes of stomatal guard cells and palisade mesophyll cells are regular enough to allow cell volume and surface area to be predicted from linear dimensions, the shapes of spongy mesophyll cells are irregular and highly lobed. As a consequence, spongy mesophyll cell volume cannot be calculated easily from a single linear dimension. To extend these analyses to the spongy mesophyll we tested whether linear cell dimensions were predicted by genome size, as has been shown for guard cell length [40]. Genome size was a strong predictor of cell diameters of stomatal guard cells, palisade mesophyll cells, and spongy mesophyll cell lobes (electronic supplementary material, table S2 and figure S3). We found no relationship between genome size and mesophyll porosity (electronic supplementary material, figures S3 and S4), which is the volumetric airspace fraction of the leaf, likely because many combinations of cell sizes and packing densities can result in the same porosity [41]. Despite the role of porosity in facilitating diffusion in the intercellular airspace [42], traits related to cellular organization within the mesophyll are likely to have a greater influence than porosity on the diffusive conductance of $CO_2$ through

Proc. R. Soc. B 288: 20203145

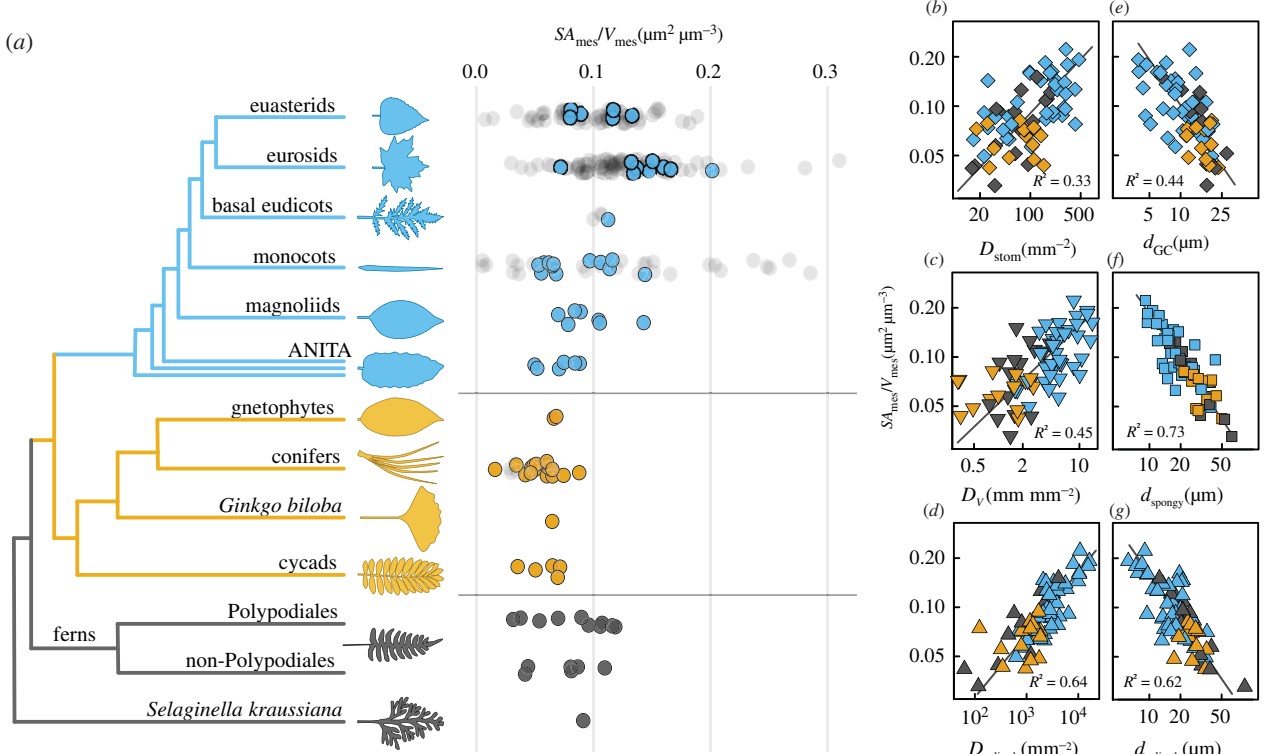

**Figure 2.** Mesophyll surface area per mesophyll volume ($SA_{mes}/V_{mes}$) scales with cell size, cell packing densities, and 2C genome size across vascular plants. (a) Distribution of $SA_{mes}/V_{mes}$ across 86 species of terrestrial vascular plants (coloured points: angiosperms, blue, top of tree; gymnosperms, orange, mid-tree; ferns and fern allies, dark grey, bottom of tree) compared to values computed from the literature (shaded grey dots, 81 angiosperms and four gymnosperms; see electronic supplementary material, Methods). Packing densities of (b) stomata on the leaf surface ($D_{stom}$), (c) veins ($D_V$), and (d) palisade mesophyll cells ($D_{palisade}$) all scaled positively with $SA_{mes}/V_{mes}$ while the diameters of (e) stomatal guard cells ($d_{GC}$), (f) spongy mesophyll cells ($d_{spongy}$), and (g) palisade mesophyll cells ($d_{palisade}$) all scaled negatively with $SA_{mes}/V_{mes}$. Solid lines represent standardized major axes. All bivariate relationships remained highly significant after accounting for shared evolutionary history (electronic supplementary material, table S2).

the intercellular airspace and into the photosynthetic mesophyll cells [33].

Because cell surfaces can be in contact with other cells and be unavailable for $CO_2$ absorption, we tested whether the effect of genome size extends beyond limiting the sizes and packing densities of cells to influencing the surface area of the mesophyll tissue exposed to the intercellular airspace ($SA_{mes}$). Genome size was a strong predictor of the total surface area per tissue volume of the mesophyll cells exposed to the intercellular airspace, $SA_{mes}/V_{mes}$ (figure 1e,f; electronic supplementary material, table S2), which is the anatomically fixed component of the leaf mesophyll that influences $CO_2$ diffusion. Our results suggest that except for a few ferns with small genomes, only angiosperms have been able to build leaves with high $SA_{mes}/V_{mes}$ (figure 2a). To explore this prediction beyond our dataset, we combined new measurements of $SA_{mes}/V_{mes}$ on the species for which we had microCT images with data extracted from the literature for 85 additional species (figure 2a; electronic supplementary material, table S3). The distribution of $SA_{mes}/V_{mes}$ among clades in our dataset was consistent with the data extracted from the literature and showed that the highest and most variable $SA_{mes}/V_{mes}$ occurs only among monocots and eudicots, suggesting that anatomical innovations among the angiosperms are responsible for the heightened $SA_{mes}/V_{mes}$ necessary to support high rates of photosynthesis. To test the prediction that genome downsizing enabled high $SA_{mes}/V_{mes}$ (figure 1e,f) via impacts on cell size and cell packing density, we tested whether $SA_{mes}/V_{mes}$ was coordinated with the sizes and packing densities of cells and tissues throughout the leaf. The packing densities of stomata,

veins, and palisade mesophyll cells were all strongly and positively related to $SA_{mes}/V_{mes}$ (figure 2b–d), while the diameters of stomatal guard cells and of spongy and palisade mesophyll cells were all strongly and negatively related to $SA_{mes}/V_{mes}$ (figure 2e–g). This whole-leaf trade-off between cell size and cell packing density (figure 1; electronic supplementary material, figure S4) was apparent in multidimensional space, in which the first axis was aligned with genome size and explained the majority of the variation whether or not phylogenetic covariation was included (electronic supplementary material, figure S5). While small genomes, small cells and high $SA_{mes}/V_{mes}$ occur predominantly among the angiosperms, some xerophytic ferns, as well as the lycophyte *Selaginella kraussiana*, also share these traits. The repeated co-occurrence of these traits among different clades and the statistically significant phylogenetic regressions between genome size, cell sizes and packing densities, and $SA_{mes}/V_{mes}$ (electronic supplementary material, table S2 and figure S5) further corroborate the role of genome size in determining the sizes and arrangement of cells and tissues throughout the leaf that enable high rates of $CO_2$ and $H_2O$ diffusion between the leaf interior and the atmosphere.

## (b) Increasing liquid-phase conductance optimizes the entire diffusive pathway

While light is intercepted primarily by the upper palisade mesophyll layer [37], $CO_2$ enters the leaf on the lower spongy mesophyll layer for most terrestrial plants, creating within the leaf opposing gradients of two of the primary reactants in

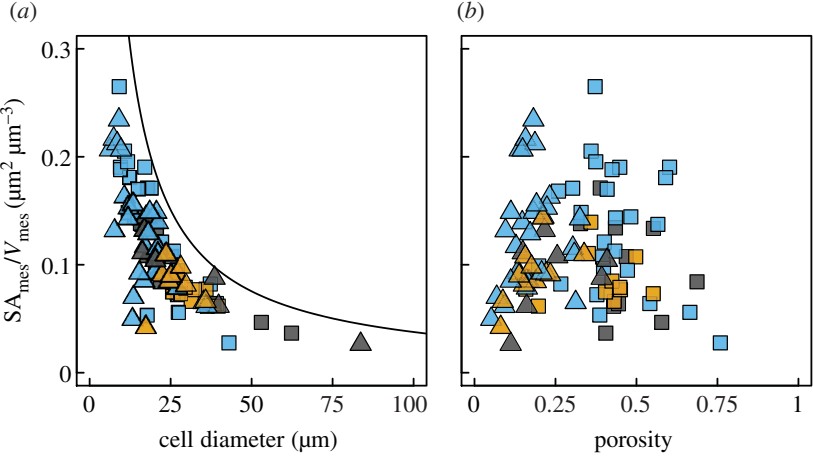

**Figure 3.** The effects of cell size and porosity on 3D mesophyll surface per mesophyll volume ($SA_{mes}/V_{mes}$). (a) Smaller cells in both the palisade (triangles) and spongy (squares) mesophyll are associated with higher $SA_{mes}/V_{mes}$. The solid line represents the theoretical maximum $SA_{mes}/V_{mes}$ calculated from the densest packing of cylinders in a rectangular volume (porosity of approx. 0.09 m$^3$ m$^{-3}$). (b) $SA_{mes}/V_{mes}$ was highest at intermediate porosity because the highest possible porosity can occur only when there are no cells and the lowest porosity occurs when all cells are in complete contact and there is no airspace. Points are coloured by plant clade, according to figure 2.

photosynthesis. Within a leaf, the spongy and palisade layers have divergent cell shapes and organizations that are thought to accommodate these opposing gradients by facilitating $CO_2$ diffusion in the gaseous and liquid-phases. Both cell size and porosity can affect $SA_{mes}/V_{mes}$ and the diffusive conductances ($g_{ias}$ and $g_{liq}$) that are considered targets of selection to increase photosynthesis [20,31,38,41,42]. To determine whether cell size or porosity has a greater effect on $SA_{mes}/V_{mes}$ and on modelled $g_{ias}$ and $g_{liq}$, we measured cell diameter, porosity, and $SA_{mes}/V_{mes}$ for the spongy and palisade layers separately for 47 species in our dataset, encompassing all major lineages of vascular plants.

The scaling of cell diameter with $SA_{mes}/V_{mes}$ (figure 2e–g) suggested that cell diameter would have a greater impact than porosity on $SA_{mes}/V_{mes}$. Smaller cells have a higher ratio of surface area to volume, an effect that could propagate up to influencing $SA_{mes}/V_{mes}$ of the entire tissue. In contrast, we predicted that porosity would not have a consistent impact on $SA_{mes}/V_{mes}$ because at very low porosities there is very little cell surface area exposed to the airspace while at very high porosities there is very little cell surface area relative to a large volume of tissue. Consistent with these predictions, decreasing cell size led to higher $SA_{mes}/V_{mes}$ across species and mesophyll layers, and variation in porosity had no consistent effect on $SA_{mes}/V_{mes}$ (figure 3). Rather, both low (less than 0.1) and high (greater than 0.6) porosities led to lower $SA_{mes}/V_{mes}$. This conditional effect of porosity on $SA_{mes}/V_{mes}$ suggests that there is a relatively narrow range of porosities that allows for simultaneous optimization of $g_{liq}$ and $g_{ias}$ in C3 plants. However, the strong and consistent effect of reducing cell size on increasing $SA_{mes}/V_{mes}$ among species and among mesophyll tissues within a leaf further implicates cell size and, by extension, genome size in controlling cell- and tissue-level traits responsible for increasing the $CO_2$ conductance of the mesophyll.

To test how these anatomical traits affect $g_{ias}$ and $g_{liq}$, we modelled $g_{ias}$ and $g_{liq}$ per unit leaf volume [24,33] as a function of cell size and porosity and compared these modelled estimates to measurements of cell diameter and mesophyll porosity taken from microCT images for the two mesophyll layers. Although this modelling did not incorporate

adjustments that can alter $g_{liq}$ over short timescales, it nonetheless shows how variation in anatomy, which is relatively fixed once a leaf has expanded [24], can influence $g_{ias}$ and $g_{liq}$. Based on simple packing of capsules, we predicted that increasing volumetric $g_{liq}$ would occur primarily by decreasing cell size, while increasing volumetric $g_{ias}$ would occur primarily by increasing porosity. We also predicted that the palisade layer, whose densely packed columnar cells channel light deep into the leaf much as a fibre optic cable directs light [37], would be optimized for $g_{liq}$ rather than for $g_{ias}$ in order to deliver $CO_2$ efficiently to the places where light is abundant. In contrast, we predicted that the spongy mesophyll layer would be optimized for high $g_{ias}$ in order to promote gaseous $CO_2$ diffusion into the upper palisade layer [23] while also scattering and absorbing light [43].

Our analysis confirmed that cell size and porosity have different effects on modelled volumetric estimates of $g_{liq}$ and $g_{ias}$ (background shading in figure 4). While increasing porosity leads to higher $g_{ias}$, it has a relatively small effect on $g_{liq}$ for a given cell size. By contrast, increasing $g_{liq}$ predominantly occurs by reducing cell size, which has only a moderate effect on $g_{ias}$ and only when porosity is relatively high. Additionally, for a given cell size, increasing porosity reduces $g_{liq}$. Thus, reductions in cell size increase both $g_{liq}$ and $g_{ias}$, but increasing porosity has opposite effects on $g_{liq}$ and $g_{ias}$. As predicted, our measurements showed that the palisade layer had lower porosities that are associated with higher $g_{liq}$, while the spongy layer had higher porosities that are associated with higher $g_{ias}$ (figure 4; electronic supplementary material, figures S12–S14). This specialization of the two layers reflects the need to maintain a high $g_{ias}$ in the spongy mesophyll where $CO_2$ is abundant to promote its diffusion into the palisade and the need to maintain high $g_{liq}$ in the palisade mesophyll where light is abundant to promote liquid-phase diffusion of $CO_2$ into the cell walls (electronic supplementary material, figures S6 and S8). Many species, particularly angiosperms, have palisade mesophyll characterized by small, highly packed cells that allow volumetric $g_{liq}$ to be higher than $g_{ias}$ of this tissue (figures 1, 4; electronic supplementary material, figure S4). This pattern suggests that $CO_2$ fixation in the palisade may be limited by the gaseous supply of $CO_2$ and not

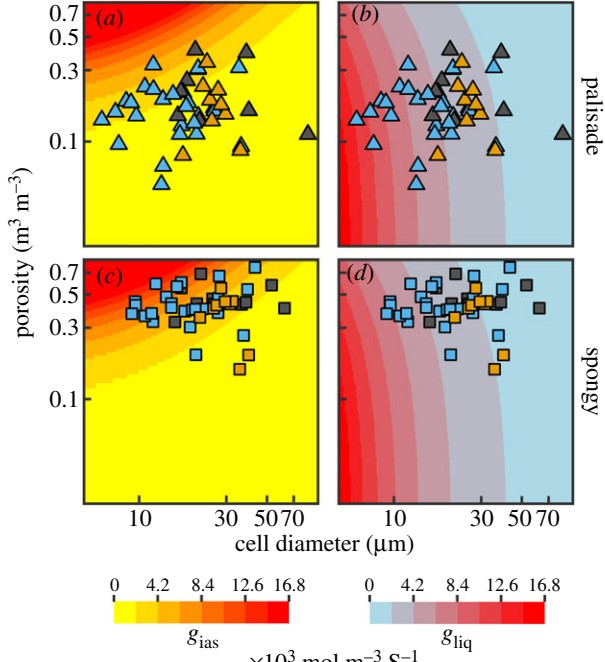

**Figure 4.** Distribution of observed cell sizes and porosities for (a,b) palisade and (c,d) spongy mesophyll relative to modelled estimates of (a,c) airspace conductance ($g_{ias}$) and (b,d) liquid-phase conductance ($g_{liq}$) to $CO_2$. Measured values of cell size and porosity (points) are plotted over theoretical conductances (coloured shading) estimated by simulating leaves of varying cell diameter and porosity (see electronic supplementary material, Methods). Points are coloured by plant clade, according to figure 2.

by its liquid-phase diffusion into cells, consistent with prior reports for hypostomatous leaves that the majority of $CO_2$ fixation occurs not at the top of the leaf where $CO_2$ is unlikely to penetrate but deeper in the palisade [43]. The structure and organization of palisade and spongy layers of the mesophyll therefore reflect the relative strengths of the opposing gradients of $CO_2$ and light.

## (c) Concluding remarks

Our results suggest that the heightened rates of leaf-level gas exchange that occurred predominantly among angiosperms are coordinated with changes not only in veins and stomata [1,5,8,9,12,13] but also in the three-dimensional organization of the leaf mesophyll tissues that limit the exchange of $CO_2$ and water. Although coordinating changes in veins, stomata, and the mesophyll undoubtedly involves multiple molecular developmental programmes, the simple scaling of genome size and cell size emerged as the predominant factor driving the increases in $SA_{mes}/V_{mes}$ and $g_{liq}$ that together enabled higher rates of $CO_2$ movement into the photosynthetic mesophyll cells. While the size and abundance of chloroplasts in the leaf will undoubtedly affect photosynthetic rates, the maximum chloroplast surface area available for $CO_2$ diffusion is limited by the surface area of the mesophyll. Because photosynthetic metabolism is the primary source of energy and matter for the biosphere, leaf-level processes are directly linked to ecological processes globally [3]. Yet theory linking ecosystem processes to organismal level metabolism has focused predominantly on the structure of vascular supply networks [44,45]. Our results suggest that the scaling of photosynthetic metabolism with resource supply networks extends beyond the vascular system and into the

photosynthetic cells of the leaf mesophyll where energy and matter are exchanged. Moreover, these results highlight the critical role of cell size in defining maximum rates of leaf gas exchange [20,46], in contrast to assumptions in current theory that terminal metabolic units are size-invariant [47,48]. Incorporating the structure of the mesophyll tissue into theory linking leaf-level and ecosystem-level processes could improve model predictions of photosynthesis. Furthermore, the physiological benefits of small cells may be one reason why the angiosperms so readily undergo genome size reductions subsequent to genome duplications [13,20,49,50]. While whole genome duplications may drive ecological and evolutionary innovation [51–53], selection for increased photosynthetic capacity subsequent to genome duplication may drive reductions in both cell size and genome size to optimize carbon fixation, reiterating a role for metabolism in genome size evolution [5,13,20].

# 3. Material and methods

## (a) Plant material

Mature, fully expanded leaves from healthy, well-watered plants were collected from greenhouses, botanical gardens, fields and other outdoor growing locations to represent a broad phylogenetic diversity of C3 vascular plants (electronic supplementary material, table S1). We chose representative angiosperms from the ANA grade, magnoliids, monocots, basal eudicots, eurosids and euasterids. We also sampled the lycophyte *Selaginella kraussiana*, 17 species of ferns from 12 families, and major groups of gymnosperms, including gnetophytes, cycads and conifers. Leaves were cut at the base of the petiole or of short stem segment, immediately put in a plastic bag with the cut end wrapped in paper towels, and scanned within 36 h of excision.

## (b) MicroCT data acquisition

MicroCT scanning was carried out at the Advanced Light Source (ALS; beamline 8.3.2; Lawrence Berkeley National Lab, Berkeley, CA, USA), the Swiss Light Source (SLS; TOMCAT Tomography beamline; Paul Scherrer Institute, Villigen, Switzerland), and the Advanced Photon Source (APS; beamline 2-BM-A,B; Argonne National Laboratory, Lemont, IL, USA). Samples were prepared less than 30 min before each scan. For laminar leaves, an approximately $1.5 \times 15$ mm piece of leaf was excised from between the midrib and the leaf outer edge. For needle and non-laminar leaves, a piece approximately 15 mm long was used. Tissue samples were enclosed between Kapton (polyimide) tape to prevent desiccation while allowing high X-ray transmittance. Samples were scanned using the continuous tomography mode capturing 1025 (ALS, APS) or 1800 (SLS) projection images at 21–25 keV, using primarily 5× (55 species; pixel size of 1.27 μm) and 10× (29 species; pixel size of 0.64 μm) objective lenses, or a 40× objective lens (2 species; pixel size of 0.1625 μm). Each scan was completed in 5–15 min.

Images were reconstructed using TomoPy [54] for all ALS samples or using the in-house reconstruction platform for SLS or APS samples. Reconstructed scans were processed using published methods [32,55], and image stacks were cropped to remove tissue that was dehydrated, damaged or contained artefacts from the imaging or reconstruction steps. The final stacks contained approximately 500–2000 eight-bit grayscale images (downsampled from 16 or 32-bit images).

## (c) Leaf trait analysis

Leaf and mesophyll thickness were measured on cross-sectional slices of the image stack. Cell diameter was measured on at

least 10 cells for each mesophyll layer on paradermal slices of the stack, as well as for guard cell length and diameter. For spongy mesophyll cells with lobed or irregular shapes, cell diameter was measured on the lobes of the cells and not on their presumed centres [56]. Some leaves had only palisade-like or spongy-like cells, resulting in some species having data for only one cell type (electronic supplementary material, table S1). To estimate cell volume, we assumed stomatal guard cells and palisade mesophyll cells were shaped as capsules with length equal to twice the diameter of the cylinder (e.g. $d_{\mathrm{palisade}}$ or $d_{\mathrm{GC}}$), allowing for cell volume to be calculated as [20]

$$V = \frac{5}{96}\pi(2d)^3.$$

We compared these estimates of mature cell volume to published measurements of meristematic cell volumes as a function of genome size [19]. We used empirical relationships between meristematic cell volume and nuclear volume and between nuclear volume and genome size [19] to estimate the relationship between meristematic cell volume and genome size, consistent with a prior analysis [20]. To estimate maximum meristematic cell packing densities in 2D, we assumed meristematic cells were shaped as spheres and calculated the maximum packing density (number of cells per area) as one divided by the cross-sectional area of the sphere, following published methods for stomata [4].

Palisade cell packing density in 2D was measured on stacks from paradermal planes through the palisade tissue by averaging per species the counts of palisade cells present within three defined areas. Stomatal density and vein density were measured on the original uncropped image stack to maximize the area measured. Scans in which stomata were difficult to discern or in which vein density would have been obviously biased (e.g. high fraction of the scan containing a higher order vein) were not measured for these traits.

To extract surface area and volumes, mesophyll cells, airspace, vasculature (combined veins and bundle sheath) and background (including the epidermis) were segmented using published methods [32,55] and ImageJ [57]. Airspace volume ($V_{\mathrm{pores}}$), mesophyll cell volume ($V_{\mathrm{cells}}$), both summing up to the total mesophyll volume ($V_{\mathrm{mes}}$), vasculature volume ($V_{\mathrm{veins}}$) and the surface area exposed to the intercellular airspace ($\mathrm{SA}_{\mathrm{mes}}$) were then extracted using published methods [32] with the ImageJ plugin BoneJ [58], or using a custom Python program [55] (https://github.com/plant-microct-tools/leaf-traits-microct). $\mathrm{SA}_{\mathrm{mes}}/V_{\mathrm{mes}}$ is less sensitive to leaf thickness than the commonly measured $S_{\mathrm{m}}$, i.e. $\mathrm{SA}_{\mathrm{mes}}$ per leaf area (electronic supplementary material, figure S8 and table S1). For separate quantification of traits from palisade and spongy mesophyll, segmented stacks were cropped at the interface between tissues or where vasculature was present, in order to accurately characterize $\mathrm{SA}_{\mathrm{mes}}$, volumes and cell diameter within those tissues.

Because our sampling included scans made at different magnifications, we tested the effect of magnification on measurements of cell size and $\mathrm{SA}_{\mathrm{mes}}$ (electronic supplementary material Results). Overall, lower magnification scans resulted in small (less than 5% for most scans) but significant changes in cell diameter and $\mathrm{SA}_{\mathrm{mes}}$ (electronic supplementary material, figure S6 and S7). However, reanalysis of scaling relationships reported in figure 2 incorporating this error showed that all relationships remained as significant as those in the original dataset (electronic supplementary material, table S3), suggesting that our results are robust to inclusion of scans with different magnifications. SMA slopes diverged only slightly between magnifications and most often were not significantly different (electronic supplementary material, table S4).

## (d) Genome size data

Existing 2C genome size (pg) data available in the Kew Plant DNA C-values Database [59] were matched to the majority of species in our dataset. Fresh leaf samples of species not in the database were collected at the University of California Botanical Garden, Berkeley CA from the same plants imaged. Genome sizes (electronic supplementary material, table S1) were measured by the Benaroya Research Institute, Virginia Mason University, using the *Zea mays* or *Vicia faba* standards and following standard protocols [60].

## (e) Simulating conductance using cell size and porosity

To model $g_{\mathrm{liq}}$ and $g_{\mathrm{ias}}$ (background shading in figure 4), we used all possible combinations of cell diameter (5–124 µm in 0.1 µm steps; 1 µm below and 40 µm above the range in our data) and porosity (0.02–0.96 in 0.01 steps; 0.03 below and 0.01 above the range in our data). For $g_{\mathrm{liq}}$, we approximated cells as capsules [39], with diameter $d$ and height $3d$, and generated the densest lattice possible, consisting of 30 cells in a $(5d)^2$ projected area (electronic supplementary material, figure S10), with a total volume of $2d \times$ projected area and a total porosity of 0.186 (see electronic supplementary material, Methods for further details). Simulating porosity above or below 0.186 was done by changing pore volume and keeping cell volume constant, which modified total lattice volume to represent either a looser cell packing or cells inflated and deformed into each other.

Liquid-phase conductance per mesophyll volume was computed [24] as a function of the surface area exposed to the intercellular airspace per volume, itself a function of cell diameter and porosity within the cell lattice, using published values for the different resistance components [24] (see electronic supplementary material). For $g_{\mathrm{ias}}$, we accounted for tortuosity and diffusive path lengthening as functions of porosity [33], and mesophyll thickness as a function of cell diameter as observed in our dataset ($R^2 = 0.21$, $p < 0.0001$; electronic supplementary material, figure S11).

## (f) Statistical analysis

All analyses, simulations and conductance computations were carried out in R 4.0.3 [61]. Standardized major axes were computed using the smatr package [62], and phylogenetic analyses (reduced major axis, generalized least-squares regression and principal component analysis) are detailed in the electronic supplementary material, Methods.

Data accessibility. Data are available as electronic supplementary material for microCT data (electronic supplementary material, table S1) and for literature data (electronic supplementary material, table S2). Code to generate the theoretical conductance values is provided as a R script. Segmented microCT images are available on Zenodo at doi:10.5281/zenodo.3606064 (https://zenodo.org/record/3606064). A preprint version of this work is available [63].

Authors' contributions. G.T.-R., J.M.E. and C.R.B. planned the project, building from ideas of C.K.B. and M.A.Z., and with contribution from C.K.B., M.A.Z. and M.E.G. G.T.-R., J.M.E., A.B.R., C.R.B., A.J.M., C.K.B., M.A.Z. and D.T. acquired microCT data. G.T.-R. and J.M.E. segmented the microCT images and extracted data from them. G.T.-R. and A.B.R. planned the analysis, analysed the data and created the simulated dataset. K.A.S. collected plant material and prepared samples for genome size analysis. D.T. contributed finite-element modelling. G.T.-R., A.B.R., K.A.S. and C.R.B. wrote the manuscript, with contributions from all authors. All authors approved the final version.

Competing interests. We declare we have no competing interests.

Funding. This work was supported by a Katherine Esau Fellowship to G.T.-R., by the Austrian Science Fund (FWF), projects M2245 and P30275, and by US NSF grant DEB-1838327. The Advanced Light Source is supported by the Director, Office of Science, Office of Basic Energy Sciences, of the US Department of Energy under Contract no. DE-AC02-05CH11231.

Acknowledgements. We thank the University of California Botanical Garden (Berkeley, CA), the UC Davis Botanical Conservatory (Davis,

CA) and the UC Davis Arboretum (Davis, CA) for plant material, and the Paul Scherrer Institute, Villigen, Switzerland for provision of synchrotron radiation beam time at beamline TOMCAT. We thank the many who collected plant material on our behalf.

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
