## [Peer Review File · Proceedings of the Royal Society B: Biological Sciences]

Review History

RSPB-2020-2176.R0 (Original submission)

Review form: Reviewer 1

Recommendation

Major revision is needed (please make suggestions in comments)

Scientific importance: Is the manuscript an original and important contribution to its field?

Excellent

General interest: Is the paper of sufficient general interest?

Good

Quality of the paper: Is the overall quality of the paper suitable?

Good

Is the length of the paper justified?

Yes

Should the paper be seen by a specialist statistical reviewer?

Yes

Do you have any concerns about statistical analyses in this paper? If so, please specify them explicitly in your report.

Yes

It is a condition of publication that authors make their supporting data, code and materials available - either as supplementary material or hosted in an external repository. Please rate, if applicable, the supporting data on the following criteria.

Is it accessible?

Yes

Is it clear?

Yes

Is it adequate?

Yes

Do you have any ethical concerns with this paper?

No

Comments to the Author

The paper addresses the relationship between genome size and structure of leaf mesophyll as the determinant of CO₂ diffusion in the leaf, making an argument for the association of small genome sizes in angiosperms with decline of CO₂ ambient concentrations. The paper uses a combination of high-tech imaging techniques of leaf structures and modelling to show how leaf structure is constrained by cell sizes and hence genome sizes. I appreciate the range of techniques used to approach the problem, but sometimes, it is not entirely clear whether a certain result comes from the model or from measurement.

Comments

li 89: this standardization (which is used extensively in the paper) assumes isometric scaling between volume and surface area of mesophyll. To what extent is this assumption reasonable? A figure in the electronic appendix may support this issue.

li 98: this is true, but in most cases the relationship between genome size and cell size is linear in log-log space, not triangular (i.e. with higher variation of cell sizes in smaller genomes).

li 113: I appreciate the phylogenetic range of taxa sampled.

li 119 and elsewhere (li 166 for example): it is not fully clear where sizes of meristematic cells come from. Are they based only (i.e. calculated from) on the measurements of genome size? Is it reasonable? More detail on the meristem cell size estimation is necessary. BTW Structures of meristematic tissues and numbers of cells involved in them differ strongly between angiosperms and monilophytes/Lycopods. Is there any evidence that the relationships know from angiosperms hold also for these groups?

li 138: is this based on modelling or on data?

li 157: "traits are having greater influence..." than what?

li 172: in two dimensions only? This should be OK for log scaling, but not for arithmetic scaling

li 188: most variable SA/V means greater flexibility in that trait, but potentially also lack of selection pressure on that trait.

li 194: I would appreciate a multivariate analysis here to show multidimensional relationships among these traits

li208 (Fig. 2): what are the grey points in monocots and eudicots?

li 210 and Table S2: there is also an option of phylogenetic major axis regression – why it has not been used? It is conceptually not fully correct to compare phylogenetic GLS and nonphylogenetic SMA – they differ both in variance-covariance matrices and in assumptions on x and y random variation.

li 370: measuring cell diameter on lobes of cells introduces a bias, which should be quantified I guess.

li 370ff: the distinction between palisade and spongy parts of the mesophyll were clear in all species?

li 407: this may also be handled by taking magnification as covariate

li 412: the GS data depend also on the ploidy level of the taxon. While species undergo reduction of GS after a polyploidization event, recent polyploidization events must be paid attention to, namely if a species has both diploid and polyploid populations/individuals. Have all species been checked against this?

Appendix: I would appreciate if the individual tables and figures are arranged consecutively – currently it is difficult to find anything there.

Appendix li 282: same values of what?

Appendix li 370: this assumes a strong phylogenetic signal, which probably does not need to be the case (see Fig. 2). ML estimation of Pagel lambda might be more appropriate for pgl.

Review form: Reviewer 2

Recommendation

Major revision is needed (please make suggestions in comments)

Scientific importance: Is the manuscript an original and important contribution to its field?

Good

General interest: Is the paper of sufficient general interest?

Excellent

Quality of the paper: Is the overall quality of the paper suitable?

Excellent

Is the length of the paper justified?

No

Should the paper be seen by a specialist statistical reviewer?

No

Do you have any concerns about statistical analyses in this paper? If so, please specify them explicitly in your report.

No

It is a condition of publication that authors make their supporting data, code and materials available - either as supplementary material or hosted in an external repository. Please rate, if applicable, the supporting data on the following criteria.

Is it accessible?

Yes

Is it clear?

Yes

Is it adequate?

Yes

Do you have any ethical concerns with this paper?

No

Comments to the Author

Theroux Rancourt and co-workers present a study of the relationship between genome size and the size and packing densities of mesophyll cells. They show that mesophyll cell size and packing density are related to the genome size and the time of divergence. They concluded that reduced genome size resulted in higher S_{Ames}/V_{mes} and increased gm. The topic is of great interest. The manuscript is well written and methodologically correct. These data are important for a fundamental understanding of the evolution of structure-function relationships of plants. My main concern is whether we can use S_{Ames}/V_{mes} as a proxy for Sc/S given that a) Sc/S_{mes} varies spatially within the mesophyll b) Sc/S_{mes} is highly variable across plant groups and also related to phylogenetic age. This should be discussed. Overall, the discussion needs to be broadened.

Decision letter (RSPB-2020-2176.R0)

16-Nov-2020

Dear Dr Th eroux-Rancourt:

I am writing to inform you that your manuscript RSPB-2020-2176 entitled "Maximum CO₂ diffusion inside leaves is limited by the scaling of cell size and genome size" has, in its current form, been rejected for publication in Proceedings B.

This action has been taken on the advice of referees, who have recommended that revisions are necessary. With this in mind we would be happy to consider a resubmission, provided the comments of the referees are fully addressed. However please note that this is not a provisional acceptance.

Sincerely,
 Professor Hans Heesterbeek
 mailto: proceedingsb@royalsociety.org

Associate Editor

Board Member: 1

Comments to Author:

This manuscript has now been seen by two external reviewers whose comments appear below. I would like to apologize for the very long time it took for our review process, but agree with both referees that the described work is exciting and that the writing is of high quality. The authors should take all reviewer comments seriously, with a particular focus on statistical/methodological issues that are raised by referee 1. I look forward to receiving an appropriately revised version of this manuscript.

Reviewer(s)' Comments to Author:

Referee: 1

Comments to the Author(s)

The paper addresses the relationship between genome size and structure of leaf mesophyll as the determinant of CO₂ diffusion in the leaf, making an argument for the association of small genome sizes in angiosperms with decline of CO₂ ambient concentrations. The paper uses a combination of high-tech imaging techniques of leaf structures and modelling to show how leaf structure is constrained by cell sizes and hence genome sizes. I appreciate the range of techniques used to approach the problem, but sometimes, it is not entirely clear whether a certain result comes from the model or from measurement.

Comments

li 89: this standardization (which is used extensively in the paper) assumes isometric scaling between volume and surface area of mesophyll. To what extent is this assumption reasonable? A figure in the electronic appendix may support this issue.

li 98: this is true, but in most cases the relationship between genome size and cell size is linear in log-log space, not triangular (i.e. with higher variation of cell sizes in smaller genomes).

li 113: I appreciate the phylogenetic range of taxa sampled.

li 119 and elsewhere (li 166 for example): it is not fully clear where sizes of meristematic cells come from. Are they based only (i.e. calculated from) on the measurements of genome size? Is it reasonable? More detail on the meristem cell size estimation is necessary. BTW Structures of meristematic tissues and numbers of cells involved in them differ strongly between angiosperms and monilophytes/lycopods. Is there any evidence that the relationships known from angiosperms hold also for these groups?

li 138: is this based on modelling or on data?

li 157: "traits are having greater influence..." than what?

li 172: in two dimensions only? This should be OK for log scaling, but not for arithmetic scaling

li 188: most variable SA/V means greater flexibility in that trait, but potentially also lack of selection pressure on that trait.

li 194: I would appreciate a multivariate analysis here to show multidimensional relationships among these traits

li208 (Fig. 2): what are the grey points in monocots and eudicots?

li 210 and Table S2: there is also an option of phylogenetic major axis regression – why it has not been used? It is conceptually not fully correct to compare phylogenetic GLS and nonphylogenetic SMA – they differ both in variance-covariance matrices and in assumptions on x and y random variation.

li 370: measuring cell diameter on lobes of cells introduces a bias, which should be quantified I guess.

li 370ff: the distinction between palisade and spongy parts of the mesophyll were clear in all species?

li 407: this may also be handled by taking magnification as covariate

li 412: the GS data depend also on the ploidy level of the taxon. While species undergo reduction of GS after a polyploidization event, recent polyploidization events must be paid attention to, namely if a species has both diploid and polyploid populations/individuals. Have all species been checked against this?

Appendix: I would appreciate if the individual tables and figures are arranged consecutively – currently it is difficult to find anything there.

Appendix li 282: same values of what?

Appendix li 370: this assumes a strong phylogenetic signal, which probably does not need to be the case (see Fig. 2). ML estimation of Pagel lambda might be more appropriate for pgl.

Referee: 2

Comments to the Author(s)

Theroux Rancourt and co-workers present a study of the relationship between genome size and the size and packing densities of mesophyll cells. They show that mesophyll cell size and packing density are related to the genome size and the time of divergence. They concluded that reduced genome size resulted in higher S_{mes}/V_{mes} and increased gm. The topic is of great interest. The manuscript is well written and methodologically correct. These data are important for a

fundamental understanding of the evolution of structure-function relationships of plants. My main concern is whether we can use S_{Ame}/V_{mes} as a proxy for S_c/S given that a) S_c/S_{mes} varies spatially within the mesophyll b) S_c/S_{mes} is highly variable across plant groups and also related to phylogenetic age. This should be discussed. Overall, the discussion needs to be broadened.

Author's Response to Decision Letter for (RSPB-2020-2176.R0)

See Appendix A.

RSPB-2020-3145.R0

Review form: Reviewer 2 (Tiina Tosens)

Recommendation

Accept as is

Scientific importance: Is the manuscript an original and important contribution to its field?

Excellent

General interest: Is the paper of sufficient general interest?

Excellent

Quality of the paper: Is the overall quality of the paper suitable?

Excellent

Is the length of the paper justified?

Yes

Should the paper be seen by a specialist statistical reviewer?

No

Do you have any concerns about statistical analyses in this paper? If so, please specify them explicitly in your report.

No

It is a condition of publication that authors make their supporting data, code and materials available - either as supplementary material or hosted in an external repository. Please rate, if applicable, the supporting data on the following criteria.

Is it accessible?

Yes

Is it clear?

Yes

Is it adequate?

Yes

Do you have any ethical concerns with this paper?

No

Comments to the Author

All my suggestions have been implemented

Decision letter (RSPB-2020-3145.R0)

21-Jan-2021

Dear Dr Th  roux-Rancourt

I am pleased to inform you that your Review manuscript RSPB-2020-3145 entitled "Maximum CO₂ diffusion inside leaves is limited by the scaling of cell size and genome size" has been accepted for publication in Proceedings B.

The referee does not recommend any further changes. Therefore, please proof-read your manuscript carefully and upload your final files for publication. Because the schedule for publication is very tight, it is a condition of publication that you submit the revised version of your manuscript within 7 days. If you do not think you will be able to meet this date please let me know immediately.

To upload your manuscript, log into <http://mc.manuscriptcentral.com/prsb> and enter your Author Centre, where you will find your manuscript title listed under "Manuscripts with Decisions." Under "Actions," click on "Create a Revision." Your manuscript number has been appended to denote a revision.

You will be unable to make your revisions on the originally submitted version of the manuscript. Instead, upload a new version through your Author Centre.

- 1) A text file of the manuscript (doc, txt, rtf or tex), including the references, tables (including captions) and figure captions. Please remove any tracked changes from the text before submission. PDF files are not an accepted format for the "Main Document".
- 2) A separate electronic file of each figure (tiff, EPS or print-quality PDF preferred). The format should be produced directly from original creation package, or original software format. Please note that PowerPoint files are not accepted.
- 3) Electronic supplementary material: this should be contained in a separate file from the main text and the file name should contain the author's name and journal name, e.g `authorname_procb_ESM_figures.pdf`

All supplementary materials accompanying an accepted article will be treated as in their final form. They will be published alongside the paper on the journal website and posted on the online figshare repository. Files on figshare will be made available approximately one week before the accompanying article so that the supplementary material can be attributed a unique DOI. Please see: <https://royalsociety.org/journals/authors/author-guidelines/>

4) Data-Sharing and data citation

It is a condition of publication that data supporting your paper are made available. Data should be made available either in the electronic supplementary material or through an appropriate

repository. Details of how to access data should be included in your paper. Please see <https://royalsociety.org/journals/ethics-policies/data-sharing-mining/> for more details.

<http://datadryad.org/submit?journalID=RSPB&manu=RSPB-2020-3145> which will take you to your unique entry in the Dryad repository.

Once again, thank you for submitting your manuscript to Proceedings B and I look forward to receiving your final version. If you have any questions at all, please do not hesitate to get in touch.

Sincerely,
Professor Hans Heesterbeek
<mailto:proceedingsb@royalsociety.org>

Associate Editor
Board Member

Comments to Author:

The authors have performed substantial additional work in response to the referees' remarks and suggestions, and thus present a more thoroughly documented study of high quality. The manuscript (incl. the Suppl. Material) reflects this additional effort, and I thank the authors for their detailed responses and attention to detail.

Reviewer(s)' Comments to Author:

Referee: 2

Comments to the Author(s).

All my suggestions have been implemented

Sincerely,
Proceedings B
<mailto:proceedingsb@royalsociety.org>

Decision letter (RSPB-2020-3145.R1)

27-Jan-2021

Dear Dr Thérroux-Rancourt

I am pleased to inform you that your manuscript entitled "Maximum CO₂ diffusion inside leaves is limited by the scaling of cell size and genome size" has been accepted for publication in Proceedings B.

You can expect to receive a proof of your article from our Production office in due course, please check your spam filter if you do not receive it. PLEASE NOTE: you will be given the exact page

length of your paper which may be different from the estimation from Editorial and you may be asked to reduce your paper if it goes over the 10 page limit.

Open Access

Paper charges

Sincerely,

Proceedings B

Appendix A

Associate Editor

Board Member: 1

Comments to Author:

This manuscript has now been seen by two external reviewers whose comments appear below. I would like to apologize for the very long time it took for our review process, but agree with both referees that the described work is exciting and that the writing is of high quality. The authors should take all reviewer comments seriously, with a particular focus on statistical/methodological issues that are raised by referee 1. I look forward to receiving an appropriately revised version of this manuscript.

We appreciate that the reviewers and editor see the time invested in this manuscript and its potential value. We appreciate the overall positive comments. We have included all statistical analyses suggested by the reviewers, and these have only bolstered our original conclusions and message. In addition, Reviewer #2 did raise an interesting point about how our results are related to other important mesophyll traits, which we address below.

Please note that in the version with tracked changes, new references have not been numbered but instead kept in Author-Date format to better track these additions and have been added at the end of the reference list. In the clean version, they are numbered, but the order will be different between the two version. Also note that the line numbers, unless otherwise mentioned, refer to the current version of the submitted manuscript.

Reviewer(s)' Comments to Author:

Referee: 1

Comments to the Author(s)

The paper addresses the relationship between genome size and structure of leaf mesophyll as the determinant of CO₂ diffusion in the leaf, making an argument for the association of small genome sizes in angiosperms with decline of CO₂ ambient concentrations. The paper uses a combination of high-tech imaging techniques of leaf structures and modelling to show how leaf structure is constrained by cell sizes and hence genome sizes. I appreciate the range of techniques used to approach the problem, but sometimes, it is not entirely clear whether a certain result comes from the model or from measurement.

We appreciate the positive feedback and apologize that we did not always make the distinction between the modeling and experimental results clear. We measured anatomical traits directly from microCT images that we have acquired, and genome size data were compiled from literature/databases with additional measurements by us (see Supplementary Table S1). These anatomical traits include linear dimensions of cells,

the volumetric porosity of the mesophyll tissue, and the surface area and volume of the mesophyll tissue. In addition, we estimated, i.e. modelled, cell volumes from the diameters of guard cells and palisade mesophyll cells by assuming a certain cell shape. Data for meristematic cells came from previously published measurements of genome size and meristematic cell volume (solid lines in Figure 1 a,b). We used these previously published empirical relationships of meristematic cell volume as a function of genome size to estimate maximum meristematic cell *packing densities* assuming a specific cell shape (solid lines in Figure 1c,d). This estimation of meristematic cell volume was identical to a prior analysis (Figure 1 in Roddy et al. 2020, reproduced below), but the extension to meristematic cell packing densities is new to the current manuscript.

We performed two types of modeling to estimate the conductances (g_{ias} and g_{liq}) to CO_2 . First, we calculated g_{liq} and g_{ias} from *published equations* while making some simplifying assumptions about certain variables—these results were presented in Figure 4. We compared these calculations of g_{liq} and g_{ias} across a range of cell sizes and porosities (colored backgrounds/isoclines in Figure 4) to our measurements of cell size and porosity performed on our microCT images (same data as in Figure 1-3 replotted as points in Figure 4). The details of these assumptions and the equations are fully outlined in the Methods and Supplemental Information. Second, we validated these analytical calculations with *finite element modeling (FEM) simulations*, which are presented solely in the Supplemental Information. Overall, these two types of modeling approaches produced similar results for the relationships between cell size, porosity, and the conductances of the intercellular airspace and of the liquid phase of the cell walls.

To better clarify the differences between measurements and modeling, we have made changes throughout the Results and Discussion section to specify when and on what cell or tissue measurements were taken, and what variables were modeled. These edits to the Results and Discussion section carry forward the last paragraph of the Introduction, which outlines what was measured and what was modeled.

On a style note, we have generally highlighted only when data were obtained from modeling, in order to avoid breaking the flow of some sentences by adding the word “measured” too often in the text. We note below some of these changes in their respective sections:

Genome downsizing enables re-organization of the leaf mesophyll

*L146: We first tested whether genome size limited the volumes and packing densities of stomatal guard cells and palisade mesophyll cells by comparing them to **published** measurements of meristematic cells as a function of genome size (Fig. 1) [19].*

Legend to Fig. 1

*L187: Minimum cell volumes (**modelled from cell diameters**) and maximum cell packing densities are limited by the size of meristematic cells (solid lines).*

Increasing liquid phase conductance optimizes the entire diffusive pathway

L254: To determine whether cell size or porosity has a greater effect on SA_{mes}/V_{mes} **and on modelled** g_{ias} and g_{liq} , we **measured** cell diameter, porosity, and SA_{mes}/V_{mes} for the spongy and palisade layers separately for 47 species in our dataset, encompassing all major lineages of vascular plants.

L287: To test how these anatomical traits affect g_{ias} and g_{liq} , we compared modelled estimates of g_{ias} and g_{liq} per unit leaf volume [24,32], in which cell size and porosity were varied independently, to measurements **of cell diameter and porosity taken from microCT images** for the two mesophyll layers.

L303: Our **analysis** confirmed that cell size and porosity have different effects on **modelled** volumetric estimates of g_{liq} and g_{ias} (background shading in Fig. 4).

Comments

li 89: this standardization (which is used extensively in the paper) assumes isometric scaling between volume and surface area of mesophyll. To what extent is this assumption reasonable? A figure in the electronic appendix may support this issue.

We agree with the reviewer that isometric scaling between cell surface area and cell volume is expected if cell shape remains the same. However, in our manuscript, SA_{mes}/V_{mes} is a tissue-level trait, where SA_{mes} is the surface of mesophyll cells exposed to the airspace (i.e. excluding the surface area of cells touching each other) and $V_{mes}=V_{cell} + V_{air}$. This distinction is important—while we would expect a scaling relationship between SA_{mes} and V_{cell} , because porosity (i.e. V_{air}) can be highly variable independent of V_{cell} , there is not any a priori relationship between SA_{mes} and V_{mes} . Thus, at a constant V_{mes} , SA_{mes} can increase or decrease depending on how densely packed the cells are (i.e. varying porosity) and by how small cells are (smaller cells have more cell surface per cell volume). To make it clear we are working on a tissue-level trait and to better define V_{mes} , we have made edits to the main text to highlight the volumes used to compute the different traits (L101):

*Because variation in leaf and mesophyll thicknesses influences SA_{mes} per leaf area [31], **expressing SA_{mes} instead by tissue volume (V_{mes} , i.e. the sum of the mesophyll cell volume, V_{cell} , and the airspace volume, V_{air})** accounts for variation in leaf construction [32,33].*

and (L108)

*Because smaller cells have a higher surface area per volume than larger cells, reducing cell size by genome downsizing would allow for **more surface area per***

cell volume (SA_{cell}/V_{cell}) and per total tissue volume (SA_{mes}/V_{mes}) that results in higher rates of CO_2 supply to the chloroplasts lining the cell walls.

While the total surface area of a cell and cell volume scale isometrically, smaller cells would enable higher SA_{mes} , but because cells in a tissue must touch each other, SA_{mes} would not necessarily scale isometrically. As the reviewer suggests, SA_{mes} and V_{cell} (not V_{mes}) are predicted to be strongly related, but only within a limit. At very high V_{cell}/V_{mes} (i.e. low porosity), adding more cells would increase V_{cell}/V_{mes} but reduce SA_{mes} because these additional cells must touch each other, thereby occluding exposed surface area. Therefore, dividing SA_{mes} by the *total volume of the tissue* (V_{mes}) does not inherently assume an isometric scaling between SA_{mes} and V_{mes} .

Below is a 2D conceptual diagram to help explain this point, showing hypothetical cross-sections of two different leaves. *Note that this example is for 2D, so 2D perimeter and 2D area are equivalent to surface area and volume in 3D, respectively.* The area occupied by the entire space (defined by the red perimeter around the green cells) is the same in both leaves. However, the two leaves differ in cell size and porosity. The leaf on the left has cells with half the radius of the leaf on the right, and the perimeter exposed to the intercellular airspace (the blue lines; P_{ias}) is much higher for the leaf with small cells (16π) than the leaf with large cells (10π), even though the total cell area (A_{cell}) is smaller than the leaf with large cells. If porosity were held constant in these two examples, then the leaf on the left with small cells would have an even higher P_{ias} because more cells would need to be added to bring A_{cell} (currently, 18π) up to the same level as the leaf on the right (28π). Importantly, all of these modifications are occurring while A_{mes} (the total area inscribed by each of the two examples) *remains the same*. Thus, dividing P_{ias}/A_{mes} does not inherently assume isometric scaling. This same principle can be extended to 3D, in which SA_{mes} (the 3D equivalent of P_{ias}) can vary as a function of cell size and porosity and *independently* of the size of the bounding box of the tissue, V_{mes} (the 3D equivalent of A_{mes} in the diagram).

As pointed out by the reviewer, isometric scaling would be expected for *single cells* that do not change shape. However, *if cell shape changes*, then SA_{cell} and V_{cell} can vary

independently. For example, if a sphere is flattened in one axis to become an ellipsoid, then its surface area increases dramatically even while its volume remains constant. This example is for shapes ranging from spheres to ellipsoids, and while the exact numbers differ for other shapes (e.g. lobed shapes of mesophyll cells; Thérroux-Rancourt et al. 2020) the same general relationships apply. However, when considering the multiple cells that form a tissue, the *tissue-level* scaling relationship between exposed SA_{mes} and V_{cell} would not necessarily be predictable because cells touch each other, preventing some of their surface area from contributing to tissue-level SA_{mes} .

li 98: this is true, but in most cases the relationship between genome size and cell size is linear in log-log space, not triangular (i.e. with higher variation of cell sizes in smaller genomes).

Yes, we agree. In previous reports, there was little or no heteroskedasticity in log-log space. However, in our recent re-analysis of many datasets, in which we calculated volumes of cells (and not only linear cell dimensions), there was heteroskedasticity, i.e. there was greater variation of cell volume among small genomes than there was among large genomes (Figure 1 in Roddy et al. 2020, below). One main point of our argument is that smaller genomes allow for greater variation in final size (see our Figure 1b and figure below), even if plants do not always take advantage of that greater range in available trait space.

Log-log relationship between genome size and cell volume; figure adapted from Roddy et al. 2020. The solid line represents the meristematic cell volumes as a function of genome size, which is the exact same line plotted in Figure 1b in the present manuscript. The difference between the figure above and Figure 1b is that in the figure above, the y-axis is in cubed units, while in Figure 1b the y-axis has been linearized by taking the cube root. The yellow arrows emphasize the greater variation—even in log-log space—of cell size when genomes are smaller.

li 113: I appreciate the phylogenetic range of taxa sampled.

We are grateful that the reviewer appreciated the range of taxa sampled. This dataset has required several years and many people to collect and analyze.

li 119 and elsewhere (li 166 for example): it is not fully clear where sizes of meristematic cells come from. Are they based only (i.e. calculated from) on the measurements of genome size? Is it reasonable? More detail on the meristem cell size estimation is necessary. BTW Structures of meristematic tissues and numbers of cells involved in them differ strongly between angiosperms and monilophytes/lycopods. Is there any evidence that the relationships known from angiosperms hold also for these groups?

We apologize that this was not completely clear in the manuscript. In line 414 (formerly 377) of the Methods we stated “We compared these estimates of mature cell volume to **published measurements of** meristematic cell volumes as a function of genome size (Simova and Herben 2012).”

The dataset of Simova and Herben was compiled from multiple previously published papers that reported independent measurements of nuclear volume, meristematic cell volume, and genome size for a variety of diploid herbaceous species. They reported two empirically derived scaling relationships:

$$\log(\text{meristematic cell volume}) \sim \log(\text{nuclear volume of shoot meristem})$$

and

$$\log(\text{nuclear volume of shoot meristem}) \sim \log(2C \text{ genome size}).$$

We used these two empirical relationships to derive the relationship

$$\log(\text{meristematic cell volume}) \sim \log(\text{genome size})$$

which is presented as the solid line in Figure 1b. This log-log relationship was then transformed into arithmetic space (by taking the antilogarithm) for plotting in Figure 1a. Because the relationship between meristematic cell volume and genome size is based on empirical measurements drawn from a variety of independent sources by Simova and Herben (2012), we are very confident in this relationship. The text in the methods has been updated to include more details about the estimation of meristematic cell volumes and packing densities as functions of genome size; it now reads (L414):

*We compared these estimates of mature cell volume to **published measurements of** meristematic cell volumes as a function of genome size [19]. **We used the empirical relationships between meristematic cell volume and nuclear volume and between nuclear volume and genome size [19] to estimate the relationship between meristematic cell volume and genome size, consistent with a prior analysis [20]. To estimate maximum meristematic cell packing densities in 2D, we assumed meristematic cells were***

*shaped as spheres and calculated the maximum packing density (number of cells per area) as one divided by the cross-sectional area of the sphere, following **published methods for stomata** [4].*

We agree with the reviewer that the “structures of meristematic tissues and numbers of cells involved in them” is highly variable among taxa and even among meristems (e.g. shoot vs root). Thus, the packing densities of meristematic cells is highly variable and depends largely on cell size and shape. In order to estimate maximum meristematic packing density (i.e. the solid lines in Figure 1c,d) we had to make simplifying assumptions about meristematic cell shape and packing, which we stated in the methods. We state (see edited text above) that “*we assumed meristematic cells were shaped as spheres and calculated the maximum packing density (number of cells per area) as one divided by the cross-sectional area of the sphere, following **published methods for stomata** [4].*” That is, the maximum number of circles that can be packed in a 2D plane is equal to $1/S$, where S = enclosed cross-sectional area. Assuming cells are shaped as spheres is the most conservative estimate for maximum cell packing—i.e. the fewest cells maximally packed would be for spherical cells. Because the packing densities of mature leaf traits reported in Figure 1c,d were on an area basis (i.e. in 2D), we were interested in estimates of maximum meristematic cell packing on an area-basis rather than on a volume-basis (i.e. 2D vs 3D).

Therefore, we agree with the reviewer that actual packings of meristematic cells are variable among meristem types and evolutionary lineages. But—importantly—our estimated maximum packing density of meristematic cells (i.e. solid lines in Figure 1c,d) should be lower for every genome size than packings that occur in actual plants. Because maximum packing is a strictly physical result of cell shape and cell size deviation from spherical cells that undoubtedly occurs among lineages would result in cell packings *higher than* those we report. Incorporating this variation would not change the main result of our analysis: namely, that mature cells are larger and less densely packed (because they are larger) than meristematic tissues. For example, in Figure 1c,d decreasing sphericity in meristematic cell shape would shift the solid line up for every genome size.

li 138: is this based on modelling or on data?

Earlier in this paragraph, the text read (slightly edited now, L150):

*The shapes of palisade mesophyll cells and stomatal guard cells can be approximated as capsules, **enabling us to calculate** cell volumes ~~can be calculated~~ from linear dimensions of length or diameter (see Methods) [20,39].*

We have also edited the text around former line 138 to better highlight the source of the data (L154):

*By reducing the size of meristematic cells, genome downsizing allows for smaller **minimum cell size** and also a greater range in mature cell size **of both stomatal guard cells and palisade mesophyll cells (Fig. 1a), consistent with prior results [13,20].** These effects of genome size on cell size were also reflected in the packing densities of guard cells and palisade mesophyll cells (Fig. 1c,d).*

To address the reviewer's comment directly here: this sentence is based on data (points) reported in Figure 1a, which shows the relationship between genome size and cell volumes of palisade mesophyll and guard cells. These cell volumes were estimated assuming a specific cell shape from linear diameter measurements of these cells. This sentence argues that because meristematic cells can be smaller if the nuclear volumes inside them are smaller, then the possible range of cell sizes is greater for smaller genomes. The solid line in Figure 1a,b is the regression of meristematic cell volume against genome size taken from measurements reported by Simova and Herben (2012).

li 157: "traits are having greater influence..." than what?

We have reworded the sentence to make it more clear we were referring to porosity. The sentence now reads (L178):

*Despite the role of porosity in facilitating diffusion in the intercellular airspace [42], ~~traits other than porosity~~ traits related to cellular organization within the mesophyll are likely to have a greater influence **than porosity** on the diffusive conductance of CO₂ through the intercellular airspace and into the photosynthetic mesophyll cells [33].*

li 172: in two dimensions only? This should be OK for log scaling, but not for arithmetic scaling

Former line 171-172 stated "Meristematic cell volume as a function of 2C genome size was taken from Šimová and Herben [19] and maximum packing density of meristematic cells calculated as the reciprocal of meristematic cell cross-sectional area." As we described above in the response to the comment on line 119, we calculated the maximum packing density of spherical meristematic cells in 2D and not 3D. Because the original empirical data from Simova and Herben (2012) were in log-log space, we calculated this maximum packing density of meristematic cells in log-log space. This maximum packing density was converted to arithmetic space by calculating the antilogarithm. Presenting data solely in log-log space can be difficult to interpret (e.g. Menge et al. 2018), and so, in order to best convey the bivariate relationships to the reader, we have presented data in both arithmetic and log-log space.

li 188: most variable SA/V means greater flexibility in that trait, but potentially also lack of selection pressure on that trait.

We agree that greater variability in SA_{mes}/V_{mes} is equivalent to greater flexibility in SA_{mes}/V_{mes} . We also agree that another explanation for this greater variation could be that this trait is not under selection. Yet another possible explanation is that selection for higher SA_{mes}/V_{mes} in order to increase photosynthetic rate is opposed by selection for lower SA_{mes}/V_{mes} in order to (for example) reduce water loss. When multiple agents of selection acting on a trait oppose each other, the trait exhibits greater variation than if the multiple agents of selection acted in the same direction (Strauss and Whittall, 2006). Thus, selection for multiple functional dimensions (e.g. limiting water loss vs. maximizing carbon gain) results in more equally fit phenotypic solutions than selection along only one functional dimension (Niklas, 1994).

While our main purpose in this manuscript is to show that reducing cell size by genome downsizing *allows for* increased SA_{mes}/V_{mes} and *enables* higher rates of CO₂ diffusion, we have tried to remain agnostic about the selective pressures causing variation in SA_{mes}/V_{mes} precisely because they may be multidimensional. One dimension driving leaf economics is likely photosynthetic rate, but there are also other dimensions, such as stress tolerance, which are beyond the scope of our manuscript. As a result, we have been careful not to say that higher SA_{mes}/V_{mes} is always selected for; rather, we have said here and elsewhere (Simonin and Roddy 2018; Roddy et al. 2020) that when maintaining high rates of photosynthesis is advantageous, then smaller cells and higher SA_{mes}/V_{mes} are likely beneficial. However, there are many environmental niches in which maintaining high rates of photosynthesis is *not* advantageous (or at least not always advantageous), such as dark understory environments and dry, water-limited conditions. In these habitats many strategies seem to coexist and are manifested in a range of cell and genome sizes (Roddy et al. 2020); species that have high instantaneous rates of metabolism whose physiological function is restricted only to the times when resource availability is highest (e.g. tropical gap species or desert perennial shrubs) coexist with species that maintain lower instantaneous rates of metabolism and can tolerate resource limitation (e.g. shade-tolerant species or desert succulents). Thus, the broad diversity of environmental conditions selects for a diversity of metabolic rates and, we predict, a diversity of SA_{mes}/V_{mes} .

li 194: I would appreciate a multivariate analysis here to show multidimensional relationships among these traits

We appreciate this suggestion and agree that we have overlooked a multivariate analysis. In the revised manuscript, we report principal component analyses with and without accounting for phylogenetic relatedness (Supplementary Figure S5). Both PCAs highlighted that most of the variation between traits (77% in non-phylogenetic PCA and

61% in phylogenetic PCA) is explained by the first PC axis, which discriminates between small cell and genome sizes in one direction and high cell packing densities and SA_{mes}/V_{mes} in the other direction. Furthermore, that genome size is almost perfectly aligned with PC1 suggests that genome size underlies the fundamental tradeoff between cell sizes and cell packing densities throughout the leaf that is the core message in our paper. We thank the reviewer for suggesting this analysis, as it reiterates our primary argument. We have added the PCA figure to the supplement (see also below), added the relevant methods in the phylogenetic analyses section of the supplements, and added the bolded statement below to the main text (L214-224):

*The packing densities of stomata, veins, and palisade mesophyll cells were all strongly and positively related to SA_{mes}/V_{mes} (Fig. 2b-d), **while** the diameters of stomatal guard cells and of spongy and palisade mesophyll cells were all strongly and negatively related to SA_{mes}/V_{mes} (Fig. 2e-g). **This whole-leaf trade-off between cell size and cell packing density (Fig. 1, S4) was apparent in multidimensional space, in which the first axis was aligned with genome size and explained the majority of the variation whether or not phylogenetic covariation was included (Supplementary Figure S5). While small genomes, small cells, and high SA_{mes}/V_{mes} occur predominantly among the angiosperms, some xerophytic ferns, as well as the lycophyte *Selaginella kraussiana*, also share these traits.***

Figure S5. Principal component analysis of the cell diameter variables, $2C$ genome size ($2C$), and cell and tissue densities per area presented in Figures 1 and 2, accounting for phylogeny (a) or not (b). Correlation biplots are presented and angles between vectors represent the correlation between variables. Percentage of total variance explained by each principal component axis is presented in parenthesis, and the first principal component axis represents the majority of the total variance when accounting for phylogenetic relatedness or not. Cell diameter variables: stomatal guard cells (d_{GC}), spongy mesophyll cells

(d_{spongy}), palisade mesophyll cells ($d_{palisade}$). Cell and tissues densities: stomatal density (D_{stom}), veins density (D_V), palisade mesophyll cells packing density ($D_{palisade}$).

li208 (Fig. 2): what are the grey points in monocots and eudicots?

The grey points are the data taken from the literature. The word 'grey' was omitted from the legend and has been edited to read:

(a) *Distribution of SA_{mes}/V_{mes} across 86 species of terrestrial vascular plants from all major clades (coloured points) and compared to values computed from the literature (shaded **grey** dots, 81 angiosperms and four gymnosperms).*

li 210 and Table S2: there is also an option of phylogenetic major axis regression – why it has not been used? It is conceptually not fully correct to compare phylogenetic GLS and nonphylogenetic SMA – they differ both in variance-covariance matrices and in assumptions on x and y random variation.

Yes, we are aware that phylogenetic GLS and nonphylogenetic SMA differ in these assumptions. Our intention in using phylogenetic GLS was to test whether after accounting for shared evolutionary history there was still a significant association between each pair of variables. We were not interested in the specific phylogenetic GLS slope.

The phylogenetic reduced major axis regression (e.g. function *phyl.RMA* in R package "phytools") is only a slope test. That is, *phyl.RMA* does not perform a hypothesis test of whether the two variables are significantly associated with each other but instead tests only whether the slope of the relationship between the two variables is significantly different from a specified slope. As suggested, we have included the phylogenetic RMA results, which compares the phylogenetic RMA slope to the non-phylogenetic SMA slope (Table S6); in other words, the phylogenetic RMA tests whether phylogeny impacts the SMA scaling slope. In the 21 bivariate relationships we tested, 10 of them had phylogenetic RMA slopes that were significantly different from the SMA slope, and in every one of these 10, the phylogenetic RMA slope was in the same direction as but steeper than the SMA slope. This comparison suggests that ignoring shared evolutionary history dampens the strength of the bivariate scaling relationships.

(For these phylogenetic RMA results, we used lambda, as per the reviewer's suggestion below, rather than a Brownian motion (BM) correlation structure. We did compare the lambda result to the BM result, and while the coefficients and test statistics differed slightly, the P-values did not. The lack of difference between lambda and BM methods is likely due to the fact that for most regressions, the estimated lambda was close to 1, which is consistent with Brownian motion model of trait evolution. Yet, even for the one

regression for which lambda was smallest (0.42), the test result did not differ from that using BM.)

In our revised SI, the phylogenetic PGLS remains in order to show whether there was statistically significant coordinated evolution of each pair of traits. Following the suggestion below, we report PGLS regressions assuming a Brownian motion correlation structure as well as PGLS regressions that incorporate a maximum likelihood estimation of Pagel's lambda. More details are below, in the response to the comment on Appendix line 370.

li 370: measuring cell diameter on lobes of cells introduces a bias, which should be quantified I guess.

For spherically shaped cells, a single diameter can describe the surface area and the volume of the cell. However, for irregularly shaped cells, there are numerous diameters or widths that could be measured, making it potentially difficult to identify the diameter most indicative of other functionally relevant (but difficult-to-measure) metrics of cell size, such as surface area and volume. Spongy mesophyll cells, as the reviewer points out, are irregularly shaped. Some co-authors from the current paper are co-authors on a manuscript looking specifically at spongy mesophyll cells (Borsuk et al., preprint). That manuscript highlights that spongy mesophyll cells are predominantly tri-lobed with the three lobes aligned in a single plane and emerging from a central body (see also Theroux-Rancourt et al., 2020). Additionally, the diameters of these lobes (or 'arms', as they are termed in the other manuscript) are strongly coordinated with the lengths of these lobes and, thus, with their surface area and volume. Because these lobes increase the surface area for CO₂ diffusion (compared to being spherical), their linear dimensions are highly relevant proxies for surface area and volume. Furthermore, these lobe lengths and diameters are strongly coordinated with guard cell length and with cell packing density, suggesting that spongy cell lobe diameter is as relevant a proxy for spongy mesophyll cell size as stomatal guard cell length is for stomatal pore area. We are now citing this preprint in the methods (L406-408).

li 370ff: the distinction between palisade and spongy parts of the mesophyll were clear in all species?

Yes, for the vast majority of species for which we measured cell diameters, spongy and palisade layers were easily distinguishable. Of the 68 species on which we measured cell diameters, 59 had clearly distinguishable palisade and spongy layers. Of the nine species for which only one cell type was defined, four did not have clearly distinguishable mesophyll layers, and the other five had anatomical peculiarities which made it difficult to clearly define a spongy mesophyll tissue even in the presence of

dichotomous dense and porous mesophyll tissues (e.g. some Bromeliaceae because of their highly porous lacunae). In the latter case, only the palisade cells were measured.

Species with no clear distinction between palisade and spongy mesophyll tissue: Calamagrostis arundacea (Poaceae); Oncidium ornithorhynchum (Orchidaceae); Tectaria moorei (Tectariaceae); Welwitschia mirabilis (Welwitschiaceae)

Species in which spongy mesophyll was difficult to define even in the presence of a dichotomous dense and porous mesophyll tissues: Bromeliaceae (Bromelia hecetioides, Guzmania zahnii, Ochagavia carnea, Pitcarnia tabulaformis), Selaginella kraussiana (Selaginellaceae).

These data for palisade and spongy cell diameter are in the Excel spreadsheet version of Supplemental Table S1.

li 407: this may also be handled by taking magnification as covariate

This is indeed a good suggestion to supplement our previous tests of the magnification effect in measurements of cell size and SA_{mes}/V_{mes} . As per the reviewer's suggestion, we have included magnification as a covariate in the SMA analysis (Supplementary Table S4). This analysis allowed comparison of the slopes of the relationships presented in Figure 2 between groups of species that were scanned at 10x and 5x (species scanned at 40x were too few for this analysis). Using the original data, i.e. not downscaled and as presented in Figure 2, all slopes were equal except for two relationships, namely with guard cell diameter (d_{spongy}) and palisade cell density (D_{pal}). The deviations in magnitude of these slopes do not affect the interpretation of our results, as our previous sensitivity analysis (Supplementary Table S3) suggested. We have added Supplementary Table S4 to present these data, as well as a paragraph explaining this new analysis. We have also edited the main text (last paragraph of the *Leaf trait analysis section* in *Material and methods*), which now reads (L448-453):

*However, reanalysis of scaling relationships reported in Fig. 2 incorporating this error showed that all relationships remained as significant as those in the original dataset (Supplementary Table S3), suggesting that our results are robust to inclusion of scans with different magnifications. **SMA slopes diverged only slightly between magnifications and most often were not significantly different (Supplementary Table S4).***

li 412: the GS data depend also on the ploidy level of the taxon. While species undergo reduction of GS after a polyploidization event, recent polyploidization events must be paid attention to, namely if a species has both diploid and polyploid populations/individuals. Have all species been checked against this?

We agree with the reviewer that unknown ploidy could be a problem in our analyses if, for example, we used genome size estimates of a diploid but anatomy was measured on a tetraploid. The effect of this error would likely be to weaken the predicted relationships; thus our conclusions would be conservative. Nonetheless, even with the possibility of this error we still find strong scaling relationships between genome size and cell sizes, packing densities, and total exposed SA_{mes}/V_{mes} . Two additional lines of evidence suggest that this problem may be minimal in our dataset.

First, we are not aware of any of the 51 species for which we used genome size data from Kew having multiple ploidies among their populations. As we stated in the Methods, we used the taxonomic information to query the Kew Plant DNA C-values Database for genome size estimates. For 15 species that were not in the Kew database but that were sampled in Berkeley, CA, we measured genome sizes from the same individuals used for microCT scanning. There were 23 additional species that we had sampled at other sites in previous years that were also not in the Kew database, but because obtaining samples of these plants was difficult, we were unable to measure genome size for them.

Second, in a prior study (Roddy et al. 2020), we compared anatomical traits of *experimentally generated* polyploids that varied in ploidy from diploids to hexaploids. (These plants were produced by using colchicine to arrest mitosis and increase ploidy. They were then backcrossed to produce intermediate ploidies. Thus, these plants were only a few generations since their genome duplication.) Guard cell length, stomatal density, and vein density all showed the same relationships as previously reported across vascular plants (Simonin and Roddy 2018). These experimentally generated polyploids also showed greater variation in anatomical traits among smaller genome sizes than among larger genome sizes, consistent with the patterns reported across vascular plants. It is important to note that even though the range of trait values for these experimentally produced polyploids is much smaller than among all vascular plants (e.g. genome size varied from only 1 to 5 pg and vein density varied from only 3 to 7 mm mm⁻²), the patterns present in the broader dataset of all vascular plants were recapitulated over the narrower range of trait values exhibited by the experimentally produced polyploids.

Appendix: I would appreciate if the individual tables and figures are arranged consecutively – currently it is difficult to find anything there.

We apologize and agree with the reviewer that the structure of the supplemental information could have been more logically arranged. We have restructured it in this revision so that it better reflects the structure and order of the main text. Further, the majority of the supplementary tables were moved to one Excel document, but the species list was kept in the Supplementary Information PDF for quick reference.

Appendix li 282: same values of what?

We apologize for this lack of clarity. The OLS and SMA regressions had the same R^2 and P-values. We have updated the text.

Appendix li 370: this assumes a strong phylogenetic signal, which probably does not need to be the case (see Fig. 2). ML estimation of Pagel lambda might be more appropriate for pgls.

We appreciate this suggestion. We have now included two versions of the PGLS, one using a Brownian motion correlation structure and one in which Pagel's lambda is estimated using maximum likelihood (as mentioned in the response to the comment on line 210 and Table S2 above). We note that the ML estimation of lambda in the PGLS can be unstable in some cases. For almost all bivariate relationships, the two methods agreed in the direction of the relationship (i.e. the sign of the slope) and the significance of the relationship. Only two of the 21 bivariate relationships differed in their significance (i.e. P-value) between the two methods (Supplementary Table S6).

We have also added a table (Supplementary Table S5), which includes five different metrics of phylogenetic signal for each trait.

Referee: 2

Comments to the Author(s)

Theroux Rancourt and co-workers present a study of the relationship between genome size and the size and packing densities of mesophyll cells. They show that mesophyll cell size and packing density are related to the genome size and the time of divergence. They concluded that reduced genome size resulted in higher $S_{\text{mes}}/V_{\text{mes}}$ and increased gm. The topic is of great interest. The manuscript is well written and methodologically correct. These data are important for a fundamental understanding of the evolution of structure-function relationships of plants.

We thank the referee for their positive and supportive comments on our manuscript.

My main concern is whether we can use $S_{\text{mes}}/V_{\text{mes}}$ as a proxy for S_c/S given that a) S_c/S_{mes} varies spatially within the mesophyll b) S_c/S_{mes} is highly variable across plant groups and also related to phylogenetic age. This should be discussed.

We agree with the reviewer that understanding S_c/S would be extremely useful in better modeling the conductance of the entire mesophyll pathway and that S_c can be highly

variable (e.g. Tosens et al. 2016, Onoda et al. 2017). Our analysis, however, focusses on the CO₂ diffusion pathway from the stomata up to the cell wall. At former line 181-182 (now L200), we were careful to point out that the effects of genome size on SA_{mes}/V_{mes} drive the “anatomically fixed component of the leaf mesophyll that influence CO₂ diffusion.” The path from the cell wall to the inside of the chloroplasts includes multiple components that also affect CO₂ diffusion (e.g. cell wall porosity), many of which can be modified over physiological timescales (e.g. membrane permeability, chloroplast positioning) (line 80). Precisely because S_c can be so variable between plants and can rapidly change (e.g. Tholen et al. 2008), we had to make some simplifying assumptions about the diffusive pathway inside the cell in order to model how tissue anatomy influences diffusion. Thus, our focus in the current manuscript was on the role of leaf construction in CO₂ diffusion, and how leaf construction, through cell size for example, has a direct influence on the total amount of chloroplastic area exposed to the cell walls, and as such sets a maximum value for S_c . We have reiterated this point in our revised manuscript in the Introduction. This now reads (L92-100):

*While **multiple membrane [24] and intracellular factors** such as carbonic anhydrase activity [28] and chloroplast positioning [29] can be actively controlled **to rapidly change** g_{liq} over short timescales, once a leaf is fully expanded, the structural determinants of g_{ias} and g_{liq} , which include the sizes and configurations of cells and airspace in the mesophyll, are thought to be relatively fixed [24,25,30]. **Of the various structural determinants of g_{liq} [30], the three-dimensional (3D) surface area of the mesophyll exposed to the intercellular airspace (SA_{mes}) is thought to be the most important because it defines the maximum amount of chloroplast surface area that can line the cell walls [26,27].***

In the Concluding remarks, we have also acknowledged the effect of S_c on gas exchange and now highlight the critical role of SA_{mes}/V_{mes} in influencing S_c (L341-348):

*Although coordinating changes in veins, stomata, and the mesophyll undoubtedly involves multiple molecular developmental programs, the scaling of genome size and cell size emerged as the predominant factor driving the increases in SA_{mes}/V_{mes} and g_{liq} that together enabled higher rates of CO₂ movement into the photosynthetic mesophyll cells. **While the size and abundance of chloroplasts in the leaf will undoubtedly affect photosynthetic rates, the maximum chloroplast surface area available for CO₂ diffusion is limited by the surface area of the mesophyll.***

Accounting for variability in S_c and cell wall thickness for example, would lead to variation in the volumetric g_{liq} estimates presented in Figure S14, which would change the pattern in their relationship with porosity, i.e. the isoclines in Figure 4. However, accounting for S_c and cell wall thickness would not change our conclusions. As an example, in order to maintain high g_{liq} , a leaf with large cells would have to substantially decrease cell wall thickness and/or line many more chloroplasts along the mesophyll

surface in order to overcome the limitations imposed by large cells (i.e. low cell surface area per unit volume). Modifying cell wall thickness, chloroplast number, and/or the total surface of chloroplasts in direct contact with the cell wall would be ways to change g_{liq} for leaves of similar cell diameter and porosity.

Nonetheless, increasing conductance of the diffusive pathway up to the cell wall (the part we have focused on here) would only be beneficial if diffusion inside the cell can also be elevated. As S_c/S scales with S_m/S within species or closely related species (e.g. Peguero-Pina et al. 2017, Veromann-Jürgensonn et al. 2020), SA_{mes}/V_{mes} will most likely scale with a volumetric equivalent of S_c , such as S_c/V_{mes} . It is, however, unlikely that leaf-area based S_c/S scales with volumetric-based SA_{mes}/V_{mes} for reasons similar to the lack of scaling with traits expressed on a mass-basis (e.g. Veromann-Jürgensonn et al. 2017).

Knowing the relationships between SA_{mes}/V_{mes} (our measurements) and S_c is very difficult and requires multiple imaging modalities. Unfortunately, 2D counts of chloroplasts are difficult to extrapolate to 3D, such that 2D estimates of S_c will more likely lead to biased 3D estimates. Further, acquiring true 3D S_c/V_{mes} is very challenging and requires special techniques not capable of high throughput (Richard Harwood, U. of Sydney, personal communication; see also Harwood et al., 2020). As such our dataset characterizes one of the anatomical traits influencing photosynthetic capacity that can be accurately acquired in 3D. We note that mesophyll porosity has long been measured, but our analyses suggest that SA_{mes}/V_{mes} is a more useful metric of mesophyll architecture than porosity.

Overall, the discussion needs to be broadened.

This comment regarding the discussion made us realize that the headings did not properly reflect how we structured the text. We have changed “Results” to “Results and Discussion” because the text in this section blended the description of the results with their discussion in a broader context. We have also changed the “Discussion” heading to “Concluding remarks” to better reflect the nature of this final section, which frames our work in the broader literatures of plant evolution, metabolic scaling, and genome size evolution. Given space limitations imposed by the journal, we cannot broaden these concluding remarks beyond these topics. If there are specific issues that the reviewer feels need to be more directly addressed at the cost of other topics already mentioned, we would be willing to try to fit them in.

References cited in the response to comments

- Borsuk AM, Roddy AB, Theroux-Rancourt G, Brodersen CR. 2019. Emergent honeycomb topology of the leaf spongy mesophyll (preprint). bioRxiv. doi:10.1101/852459.
- Harwood R, Goodman E, Gudmundsdottir M, Huynh M, Musulin Q, Song M, Barbour MM. 2020. Cell and chloroplast anatomical features are poorly estimated from 2D cross-sections. *New Phytologist* 225: 2567–2578.
- Menge DNL, MacPherson AC, Bytnerowicz TA, Quebbeman AW, Schwartz NB, Taylor BN, Wolf AA. 2018. Logarithmic scales in ecological data presentation may cause misinterpretation. *Nature ecology & evolution* 2: 1393–1402.
- Niklas KJ. 1994. Morphological evolution through complex domains of fitness. *Proceedings of the National Academy of Sciences* 91: 6772–6779.
- Onoda Y, Wright IJ, Evans JR, Hikosaka K, Kitajima K, Niinemets Ü, Poorter H, Tosens T, Westoby M. 2017. Physiological and structural tradeoffs underlying the leaf economics spectrum. *New Phytologist* 113: 4098.
- Peguero-Pina JJ, Sisó S, Flexas J, Galmés J, García-Nogales A, Niinemets Ü, Sancho-Knapik D, Saz MÁ, Gil-Pelegrín E. 2017. Cell-level anatomical characteristics explain high mesophyll conductance and photosynthetic capacity in sclerophyllous Mediterranean oaks. *New Phytologist* 214: 585–596.
- Roddy AB, Theroux-Rancourt G, Abbo T, Benedetti JW, Brodersen CR, Castro M, Castro S, Gilbride AB, Jensen B, Jiang G-F, et al. 2020. The scaling of genome size and cell size limits maximum rates of photosynthesis with implications for ecological strategies. *International Journal of Plant Sciences* 181: 75–87.
- Simonin KA, Roddy AB. 2018. Genome downsizing, physiological novelty, and the global dominance of flowering plants. (A Tanentzap, Ed.). *PLoS biology* 16: e2003706.
- Šímová I, Herben T. 2012. Geometrical constraints in the scaling relationships between genome size, cell size and cell cycle length in herbaceous plants. *Proceedings. Biological sciences* 279: 867–875.
- Strauss SY, Whittall JB. 2006. Non-pollinator agents of selection on floral traits. In 'Ecology and evolution of flowers'. (Eds LD Harder, SCH Barrett) pp. 120–138.
- Theroux-Rancourt G, Voggeneder K, Tholen D. 2020. Shape matters: the pitfalls of analyzing mesophyll anatomy. *New Phytologist* 225: 2239–2242.

Tholen D, Boom C, Noguchi KO, Ueda S, Katase T, Terashima I. 2008. The chloroplast avoidance response decreases internal conductance to CO₂ diffusion in *Arabidopsis thaliana* leaves. *Plant, Cell & Environment* 31: 1688–1700.

Tosens T, Nishida K, Gago J, Coopman RE, Cabrera HM, Carriquí M, Laanisto L, Morales L, Nadal M, Rojas R, et al. 2016. The photosynthetic capacity in 35 ferns and fern allies: mesophyll CO₂ diffusion as a key trait. *New Phytologist* 209: 1576–1590.

Veromann-Jürgenson L-L, Tosens T, Laanisto L, Niinemets Ü. 2017. Extremely thick cell walls and low mesophyll conductance: welcome to the world of ancient living! *Journal of Experimental Botany* 68: 1639–1653.

Veromann-Jürgenson L-L, Brodribb TJ, Niinemets Ü, Tosens T. 2020. Variability in the chloroplast area lining the intercellular airspace and cell walls drives mesophyll conductance in gymnosperms. *Journal of Experimental Botany* 71: 4958–4971.